# Molecular Mechanisms of Regulation of Root Development by Plant Peptides

**DOI:** 10.3390/plants12061320

**Published:** 2023-03-14

**Authors:** Larisa I. Fedoreyeva

**Affiliations:** All-Russia Research Institute of Agricultural Biotechnology, Timiryazevskaya 42, 127550 Moscow, Russia; fedlara@inbox.ru

**Keywords:** plant peptide, root development, CLE, RGF, RALF, molecular mechanism

## Abstract

Peptides perform many functions, participating in the regulation of cell differentiation, regulating plant growth and development, and also involved in the response to stress factors and in antimicrobial defense. Peptides are an important class biomolecules for intercellular communication and in the transmission of various signals. The intercellular communication system based on the ligand-receptor bond is one of the most important molecular bases for creating complex multicellular organisms. Peptide-mediated intercellular communication plays a critical role in the coordination and determination of cellular functions in plants. The intercellular communication system based on the receptor-ligand is one of the most important molecular foundations for creating complex multicellular organisms. Peptide-mediated intercellular communication plays a critical role in the coordination and determination of cellular functions in plants. The identification of peptide hormones, their interaction with receptors, and the molecular mechanisms of peptide functioning are important for understanding the mechanisms of both intercellular communications and for regulating plant development. In this review, we drew attention to some peptides involved in the regulation of root development, which implement this regulation by the mechanism of a negative feedback loop.

## 1. Introduction

Peptides are small molecules containing from 2 to 100 amino acids. They are involved in the regulation of cell differentiation, growth, development and defense of plants [1,2], in addition, peptides are involved in intercellular communication and signaling over long distances [3,4,5]. For a long time, peptides were not shown due attention in the life of plants. The lack of interest in the study of peptides was mainly due to their low concentrations in plants and the difficulty of their identification. Only in the early 1990s, with the description of systemin, which are peptide hormones, did interest grow as this was discovered [6].

An endogenous hydroxyproline-rich peptide, systemin, was isolated from tomato culture medium and its 18 amino acid sequence was determined [7]. The synthesized identical peptide was capable of inducing the synthesis of two proteins inhibitors of wound-induced proteinases at very low concentrations. The role of systemin as a wound hormone has been proven. Systemin is formed from prosystemin, which is a large precursor protein, as a result of hydrolytic processing, similar to animal polypeptide hormones [8]. The discovery of systemin became the basis for the discovery of a whole class of peptide hormones. Plant peptide hormones play a key role in various processes, such as cell proliferation and differentiation, stomatal development, self-incompatibility, and defense responses [9,10].

Specific receptors located on the surface of target cells recognize peptides. The formed peptide hormone-receptor complexes activate the intracellular signaling pathway through kinases and transcription factors, which then initiate various cellular processes. The molecular basis of intercellular ligand-receptor bonds is essential for the functioning of multicellular organisms. Recent advances in molecular biological analysis have made it possible to identify phytohormone receptors and signaling mediators [5]. Research in the field of studying the functions of peptides revealed their participation in many aspects of the plant life cycle, including development and environmental reactions similar to the functions of canonical phytohormones. Based on this knowledge, it is becoming increasingly common that these peptide hormones are the main regulators in plants. Peptide hormones are usually expressed and secreted in certain cells or tissues, then undergo further post-translational modifications that are necessary to perform their functions, and then are transported to target cells [11,12,13].

Previously, the greatest attention was paid to peptides with antimicrobial activity. Plant antimicrobial peptides (AMPs) are active participants in the protective barrier system of plants. They have been isolated from all organs of various plant species and have activity against phytopathogens, as well as bacteria pathogenic for humans [14]. Most natural antimicrobial peptides are 10 to 50 amino acids in length with m.m. 2 to 9 kDa, they are positively charged, have a high content of hydrophobic amino acids, and often have a helical structure. Microorganisms or their products induce either *AMP* gene expression or transcription [15,16]. AMPs can be divided into two classes based on their mechanism of action. The first group includes AMPs, which, interacting with the phospholipids of pathogens, make their membranes permeable, resulting in the death of pathogenic microorganisms. Another group includes pathogen cell-penetrating peptides (CPPs), which are capable of introducing various toxic substances into cells through interactions with membrane phospholipids. AMPs and CPPs are part of the host’s nonspecific defense system and are active against various types of microorganisms [17,18,19,20].

## 2. Classification of Peptide Hormones

Peptides are localized in all plant organs. Depending on the localization, the peptides exhibit different specific functional activities. Some of these peptides act as short local signaling molecules during plant growth and development, while others are active in environmental adaptation, acting over long distances from root to shoot. The number of peptide hormones functioning in plants is significant, and although the number of characterized peptide hormones currently exceeds the number of classical plant hormones, this is only a small part of the existing ones [21,22,23].

With the increase in the discovery of new peptide hormones, the question arose of their classification. There are several approaches to their classification: by origin, structure, post-translational modifications, localization, and functional role [11]. The most common classification of peptides is based on the structural characteristics derived from primary sequences. According to this classification, secreted peptide hormones are divided into three groups: the first group is peptides that include complex post-translational modifications followed by extensive proteolytic processing; the second is peptides with the formation of an intramolecular disulfide bond with subsequent proteolytic processing; and the third is peptides with multiple intramolecular disulfide bonds without proteolytic treatment. The first group of peptides is called small post-translationally modified peptides; the peptides of the last two groups are defined as cysteine-rich peptides (Figure 1).

Short peptides can be divided into secreted and non-secreted ones [9]. Genes for secreted peptide hormones are initially translated as biologically inactive prepropeptides, followed by the deletion of the N-terminal signal peptide. The removal of the signal peptide occurs with the help of a signal peptidase, resulting in the formation of a propeptide (Figure 1). After the removal of the signal peptide, propeptides can be divided into two groups based on the structural characteristics derived from their biogenesis pathways: propeptides that undergo proteolysis (1) and large cis-rich peptides without proteolysis (2). According to species characteristics, the peptides of the first group can be divided into two subgroups, one group being short peptides consisting of 5–20 amino acids formed after proteolytic treatment (Figure 1). The peptides of this group contain active amino groups such as Pro, Tyr, Gly or Lys, which often undergo post-translational modifications. Post-translational modifications, including proline hydroxylation or glycosylation (mainly arabinosylation) and the sulfation of tyrosine residues [20], are often important for the maturation of these peptides and play a role in stability, activity, and interaction with the receptor [24,25,26]. It is believed that mature peptides are encoded near the C-terminal region of the precursor. It has been observed that the amino acid sequences corresponding to the mature peptide motif are highly conserved within a plant family. In contrast to the mature peptide domain, other domains in the propeptide showed high amino acid sequence variability.

Cys-rich peptides of the second group can contain from 2 to 16 Cys residues and have a relatively fixed structure due to the formation of intramolecular disulfide bridges [3,27,28]. Previously, it was believed that Cys-rich peptides function in plants mainly as antimicrobial compounds upon infection by pathogenic microorganisms [11,29]. However, it was further discovered that these peptides may play an important role in stomatal structure and density, symbiosis, and a wide range of reproductive processes such as tube pollen germination, induction and rupture, gamete activation, and seed development [30,31,32,33]. Consistent with their role in reproduction, Cys-rich peptides are widely present in both female and male gametophytes, in contrast to post-translationally modified peptides, which are predominantly found in vegetative tissues.

Although most of the described plant peptides were derived from non-functional precursors, there are data indicating the existence of peptides derived from functionally active proteins that exhibit a dual function [34,35,36]. Such dual functional proteins can be the source of a functional peptide after proteolytic processing.

There is evidence that peptides can also be synthesized directly from a short open reading frame (sORF) (<100 codons) that is present in the 5′ leader sequence of mRNA, in microRNA (miRNA) primary transcripts, or in other non-coding transcripts of longer (>100 amino acids) proteins [37,38]. A feature of the transcripts encoding this type of peptide is their crossing over with long noncoding RNA transcripts, which may contain one or more ORFs [39]. An in silico database analysis of the complete genome of Arabidopsis identified a number of genes encoding small secreted peptides with ORFs ranging from 50 to 150 amino acids [40].

Peptide hormones have been found and characterized in all plant organs. In Arabidopsis thaliana, more than 1000 genes encoding secreted peptides with a potential signal function have been found in the complete genome sequence [11,41,42], but so far the molecular mechanisms controlling biogenesis and the functional role of these peptides have been studied only in a seldomly.

## 3. Zonal Localization in the Root of CLE, RGF and RALF1 Peptides

In this review, we will focus in more detail on peptides localized in roots and involved in the differentiation and regulation of root and root hair growth. Root development occurs in three main zones (Figure 2): this is the meristematic zone (MZ), in which cells actively proliferate, then the EZ zone is the zone of elongation, where cells stop proliferating and their elongation occurs, and, finally, there is the DZ zone of differentiation, where cell differentiation occurs and their fate is determined. The control of maintaining a balance between the proliferation and differentiation of stem cells, which determines the rate of root growth, is carried out not only by phytohormones, but is also modulated by peptide hormones.

In the shoot apical meristem (SAM) and root meristem (RAM) there is a specialized cellular zone—a niche of stem cells [43,44]. Stem cells are self-sustaining and produce founder cells that are capable of differentiating into organ and tissue cells. The organization of the stem cell niche in SAM and RAM differs, however, the factors involved in the regulation of stem cell maintenance are conserved in both meristems. Stem cell identification is regulated by signals from the organizing center (OC) in the SAM and the quiescent center (QC) in the RAM. Control over the zone of stem cells in the meristem is carried out by two adjacent groups of cells located in the central zone (CZ) at the top of the meristem. One group of cells is slowly dividing stem cells, and the other, located below the organizing center (OC), during the period of stem cell division, moves laterally to the peripheral zone (PZ), where they differentiate [45,46,47]. Thus, in the quiescent center (QC) in the RAM, control over the specification of stem cells and maintenance of the undifferentiated state of stem cell initials is carried out [48].

## 4. CLE Peptides Regulating the Stem Cell Niche

### 4.1. CLE Peptides and Their Localization

It is believed that the largest group of peptides identified to date is CLE, named after a family of genes associated with the CLAVATA/embryonic region [49]. The Arabidopsis genome contains 32 CLE genes. CLE genes have been found to be expressed in almost all tissues, indicating their broad biological functions [50,51]. It has been established that all CLE peptides are formed as a result of the hydrolytic processing of larger precursor proteins [52]. CLE precursor proteins contain a signal peptide at the N-terminus and a conserved 14-amino acid motif (CLE) at the C-terminus, from which the mature CLE peptide is formed as a result of post-translational modifications [53]. Accumulating evidence indicates that secreted peptides commonly undergo modifications such as proline hydroxylation, hydroxyproline arabinosylation, and tyrosine sulfation [54,55]. The CLE18 peptide was found, which differed from other peptides of the CLE family in that its CLE motif is located in the variable region of the precursor protein, and not at the C-terminus, as usual. In addition, this peptide exhibited reverse functional activity, inducing the long root phenotype [56], as compared to the synthetic 12-aa-peptide, which suppressed root elongation [57].

CLE peptides are involved in many cellular processes, including the control of apical meristem and cambium activity in shoots and roots, vascular tissue differentiation, the formation of lateral roots and nodules, early embryogenesis, stomatal development, and the response to a number of environmental factors (Figure 3) [10,58,59].

### 4.2. CLV3 Peptide Forms a Negative Feedback Loop CLV3-CLV1-WUS in RAM

CLE peptides are secreted into the extracellular space and interact with kinase receptors rich in leucine repeats (LRR-RLK) CRINKLY4 located on plasma membranes [60]. Secreted peptides are signaling molecules and are involved in intercellular communication in plants by inducing signaling pathways. As a result of intercellular communication, peptides contribute to the determination of the identity and activity of neighboring cells and the functional regulation of plant tissues and organs. The CLE-receptor peptide complexes trigger a signaling cascade whose targets are homeodomain-containing transcription factors of the *WOX* family, which regulate the maintenance of stem cell niches in plants [61]. As a result of signaling from the cells organizing the QC, in which the TF *WUS* homeodomain is localized, the identity of stem cells is established [62]. The *WOX* gene family (Wuschel-like homeobox WUS) belongs to the HOMEODOMAIN family of plant-specific transcription factors involved in various cellular processes [63]. *WOX* genes have been shown to play a crucial role in stem cell regulation [64], embryo patterning [65], and flower development [66]. The *WUS* genes are expressed in a small group of cells located in the QC [66,67]. Loss of the *WUS* function leads to stem cell differentiation [67]. On the contrary, the *WUS* expression in budding vegetative organs induces the identity of ectopic stem cells [68]. These data indicate that a strict control over the expression of *WUS* genes is necessary to maintain the required number of stem cells.

The genetic mechanism of homeostasis regulation was studied in *Arabidopsis thaliana* [69,70,71]. The *WUS* expression was found to be negatively regulated by the CLAVATA signaling pathway (CLV), which consists of a signal peptide formed after the proteolytic processing of the CLV3 protein and its receptors rich in CLV1 leucine repeats [43,62,72]. On the other hand, it has been noted that *WUS* promotes the expression of CLV3 localized in the region of stem cells. CLV3 is a signal peptide that regulates the fate of stem cells in the *Arabidopsis* apical meristem [43]. This WUS-CLV3 negative feedback loop in the stem cell niche is the basis for maintaining the stem cell zone (Figure 4).

CLV3 is a small protein that includes a conserved 13 amino acid CLE domain [62]. In mature CLV3, the seventh Hyp residue of the CLE domain is posttranslationally modified with three L-arabinose residues [73]. The arabilization of the CLV3 peptide promotes an increase in affinity for the CLV1 receptor compared to the non-arabinosylated form. The proposed role of arabinosylation is that the arabized tail distorts the conformation of the peptide backbone, which leads to a significant increase in affinity for certain receptor motifs [73].

The signaling pathway involving CLV3 has been extensively studied in several plant species and has been shown to be critical for stem cell homeostasis in shoot and flower meristems [74,75]. Moving between stem and neighboring cells, the CLV3 protein is involved in intercellular communication. Such communication between stem cells, in which the CLV3 protein is involved in the suppression of the *WUS* expression, and their neighbors is essential for stem cell homeostasis. It has been shown that an excess amount of stem cells can indirectly inhibit their daughter cells laterally, resulting in the initiation of their differentiation.

By secreting into the extracellular space, CLV3 interacts with receptor complexes consisting of CLV1, CORYNE-CLV2, and RE-CEPTOR-LIKE PROTEIN KINASE2 [76,77,78]. CLV1 activation includes autophosphorylation, interaction with membrane-bound and cytosolic kinases and phosphatases [79,80]. The CLV1 transmembrane receptor is a leucine-rich LRR in the kinase domain that is localized in the OC in the SAM [81]. As a result of the overexpression of *CLV1* in the center of the shoot meristem, there is a restriction of the transit of CLV3 from stem cells. The restriction of CLV3 transit leads to the activation of *WUS* in the OC and ensures the continuous production of stem cells and meristem activity. This regulated movement of the secreted CLV3 signal peptide allows the shoot meristem to determine the start of cell differentiation in the periphery while maintaining a stable stem cell niche in the OC [75].

The suppression of the *WUS* activity and an increase of the CLV3 activity leads to the proliferation of stem cells and a decrease in their pool. However, the loss of cells from the PZ due to the formation of lateral organs requires a compensated expansion of the stem cell niche [62,64].

### 4.3. CLE40 Peptide Forms a Negative Feedback Loop CLE40-AcR4-WOX5 in RAM

The receptor-like kinase ACR4 (ARABIDOPSIS CRINKLY4) is involved in determining the fate of pericycle cells during lateral root initiation [82]. ACR4 kinase has been determined to be involved in repression of pericycle cell division and thus controls lateral root patterning. ACR4 has also been reported to be involved in the regulation of stem cell fate in RAM through the CLE40-ACR4-WOX5 signaling loop [83] (Figure 4). The maintenance of the stem cell pool is necessary for the formation of the final root architecture, since the functioning of the root apical meristems and the formation of de novo lateral roots are completely dependent on this. In the root tip meristem, ACR4 controls the activity of cell proliferation in the columella cell line and is a key factor in both stimulating and limiting the number of cell divisions formed after the onset of organogenesis. Thus, the function of ACR4 reveals a common mechanism for the control of formative cell divisions in the meristem of the tip of the main root and during lateral root initiation. The CLE40 peptide is expressed in differentiating root stem cells. A decrease in *CLE40* expression levels delays differentiation and promotes stem cell proliferation. Conversely, elevated CLE40 levels drastically alter the *WOX5* expression domain and promote stem cell differentiation. It was found that the regulation of *WOX5* expression occurs through the interaction of the CLE40 peptide with the ACR4 kinase, since the CLE40 peptide is able to influence the expression of *ACR4.* Thus, the CLE40 peptide in differentiating cells initiates a negative feedback signal, acting through the ACR4 kinase receptor, regulating the *WOX5* expression. Although there are parallel mechanisms of regulation of the stem cell niche by the CLE40 peptide in shoots and roots, significant differences have also been identified.

### 4.4. CLE40 Peptide Forms a Negative Feedback Loop of CLE40-BAM1-WUS in SAM

The function of the CLE40 peptide in the development of *Arabidopsis* shoots was analyzed. The *WUS* expression in OC was found to be positively regulated through the CLE40-BAM1 signaling pathway. In addition to the CLV1 kinase receptor, it has been found that other receptors, such as RECEPTOR-LIKE PROTEIN KINASE2 (RPK2), CLAVATA2-CORYNE heteromer (CLV2-CRN), and BAR MERISTEM 1-3 (BAM1-3) [78,84,85,86]. Although BAM receptors share high sequence homology with CLV1, they may be involved in other functions, than CLV1. For example, *bam1* 1 and *bam2* double mutants have smaller shoot and flower meristems, thus showing the opposite phenotype to *clv1* mutants [85,86,87]. At the same time, as a result of experiments on the ectopic expression, it was found that CLV1 and BAM1 can perform similar functions in the control of stem cells [88]. It is important to note that *CLE40* is expressed in PZ differentiating cells and is limited to meristematic tissues, but not to organ-forming sites in the CZ with high *WUS* activity. Studies of *clv3* mutants with extended stem cell domains and the analysis of *wus* mutants have shown that the *CLE40* expression, in contrast to *CLV3*, is negatively controlled in a WUS-dependent manner [78] (Figure 4). In addition, it was noted that the number of cells in which *WUS* expression occurs is significantly reduced in mutant *cle40*, which indicates a positive effect of the CLE40 peptide on the size of the shoot meristem.

Thus, two antagonistic pathways CLV3-CLV1 and CLE40-BAM1 were identified that regulate the *WUS* activity in shoots. It has been suggested that BAM1, when suppressing CLV1, may have a dual function: to suppress the *WUS* expression in response to CLV3 in OC and simultaneously stimulate the *WUS* expression in response to CLE40 [84,89]. It has been suggested that there may be some cellular signal that is able to signal from CLE40-BAM1 in PZ to the meristem center to stimulate the *WUS* expression, since *WUS* is not expressed in the same cells as BAM1 [90]. As a result of signaling from CLE40-BAM1, the necessary feedback signal is provided to stimulate the activity of stem cells and thus initiate the formation of new organs.

## 5. RGF Peptides Regulating the Root Development

### 5.1. Peptide Containing Sulfated Tyrosine 1. RGF Peptides

A group of peptides responsible for maintaining the root meristem associated with the CLE peptide family in Arabidopsis was identified using various search strategies. This family of peptides was originally named ROOT MERISTEM GROWTH FACTORS (RGFs) [55]. Because of the CLEL motifs at their C-terminus, these peptides are also called CLEL or GOLVEN (GLV) [91,92]. The RGF peptide family in Arabidopsis includes 11 members, more than half of which are specifically expressed in resting central cells, columella stem cells, and the innermost layer of columella central cells at the root apex [93]. It was found that the synthesized RGF peptides are able to restore the size of the meristem by increasing the number of meristematic cells, which is accompanied by the restoration of the stem cell function in the roots of the ***tpst-****1* mutant [51,91]. On the contrary, the triple mutation ***rgf1-1, rgf2-1, rgf3-1*** leads to a decrease in the number of meristematic cells and causes a short root phenotype.

RGF precursor proteins, such as CLE precursor proteins, contain two main domains: the N-terminal signal peptide and the C-terminal RGF peptide domain [91]. Most mature RGF peptides have sulfated Tyr residues at position 2 and hydroxylated Pro residues. RGF peptides share the Asp-Tyr amino acid pair with other known sulfotyrosine peptides such as phytosulfokine (PSK) and peptide (PSY1) [94,95]. Probably, conserved regions are responsible for the main functions of the RGF peptide family, while other regions are responsible for their receptor affinity.

Three tyrosine–sulfated peptide hormones PSK, PSY1 and RGF1 have been discovered and well characterized. The peptide was identified as a growth stimulating factor in plant cell cultures [96]. This growth factor was purified and named PSK. PSK is formed from a ~80 mer precursor peptide after post-translational TPST sulfation and proteolytic processing [96,97]. The secreted PSK pentapeptide contains two sulfated tyrosines. Biochemical analysis shows that PSK binds to the leucine-rich repeat receptor kinase (LRR-RK), PSKR1 [98]. The destruction of PSKR1 in ***Arabidopsis*** has been shown to cause pleiotropic growth defects such as short roots, small leaves, and early senescence [94,99].

The second sulfated peptide is PSY1, an 18-membered secreted glycopeptide containing one sulfated tyrosine residue [94]. The sulfated PSY1 peptide has been found to be expressed in various tissues of *Arabidopsis* and, at very low concentrations, promotes cell proliferation.

The third sulfated peptide, RGF1, is a secreted 13 amino acid peptide involved in maintaining the *Arabidopsis* root stem cell niche [51]. RGFs are generated from precursor peptides (≈100 amino acids) by post-translational sulfation followed by proteolytic processing. As a result of the search for peptides responsible for the repair of defects in the root meristem, the sulfated RGF1 peptide was found and identified in the *tpst-1* mutant. The identification of RGF1 was also confirmed by the in silico screening of genes encoding sulfated peptides. This approach is based on the assumption that *tpst-1* mutant phenotypes are deficient in the biosynthesis of all functional tyrosine sulfated peptides. In addition, the functional role of the RGF1 peptide has also been established by practical biological analysis using synthetic sulfated peptides.

The main function of RGF peptides is the regulation of the development of the root system, which is carried out through an interaction with the proteins of transcription factors PLETHORA (PLT). PLT proteins are expressed in the root meristem and are involved in the formation of the root stem cell niche [100]. RGF family peptides are predominantly expressed in the stem cell region and in the innermost layer of the columella central cells and then diffuse into the meristematic region through the apoplast.

It has been noted that some members of the RGF family are also expressed in shoots reproductive organs [98,101], indicating that RGF functions diversify during plant evolution. These comprehensive functional studies of RGF peptides were carried out mainly on *Arabidopsis thaliana*.

Studies have shown that some CLAVATA3/ESR-related (CLE) and C-TERMINAL PEPTIDE PROTEINS (CEP) [102,103] contain only one RGF domain. Many of the RGF domain sequences are shared between proteins from different plant species. On the other hand, some RGF domain sequences may be unique to eudicots or monocots. For example, an RGF domain sequence (DYAQPHRKPPIHN) has been identified in six eudicot proteins, while RGF domain proteins (DYYGASVHEPRHH) have been found in four monocots. These data suggest that some RGF domain sequences may be unique to eudicots or monocots. In addition, RGF proteins contain some similar motifs, and these motifs may also play an important role in RGF function.

### 5.2. RGF1 Peptide Participate in Forming Stem Cell Niche

The root apical meristem, located at the root tip, plays a vital role in the regulation of root structure formation and adaptation to environmental stimuli [9,10]. RGFs are important for maintaining the root stem cell niche. The following scheme of participation of RGFs in maintaining the niche of root stem cells was proposed (Figure 5). RGFs peptides after proteolytic processing of their precursor proteins mature as a result of catalysis by tyrosyl protein sulfotransferase (TPST), which sulfates the tyrosine residue required for the biological activity of RGFs [55]. It has been noted that TPST mutations result in disruption of root stem cell niche maintenance, decreased meristem activity, and stunted root growth. A relationship has been found between auxin-regulated *TPST* expression, TPST mutation, and auxin distribution. The TPST mutation disrupts basal and auxin-induced expression of stem cell transcription factor *PLT* genes. At the same time, the overexpression of *PLT2* leads to the restoration of defects in the root meristem of the *tpst* mutant.

The importance of tyrosine sulfation has been noted in many studies [104,105], demonstrating a 185-fold decrease in the affinity of RGF for the RGFR1 receptor. The RxGG motif in RGFR has been identified and is responsible for the specific recognition of the sulfate group. However, some studies report the activity of the unmodified peptide, which raises the question of how important this modification really is [91,106,107]. The controversial need for the sulfation of tyrosine residues for biological activity has been demonstrated in a member of the RGF family, RGF1/CLEL8/GLV1 [96]. It has been shown that the CLEL8 peptide (CLE-like) without sulfation and hydroxylation changes the direction of root growth and the development of lateral roots, while the sulfated RGF1 peptide is able to restore the defective *tpst-1* root apical meristem [55]. In addition to increasing the specificity and affinity of peptides for their receptors, tyrosine sulfation and proline hydroxylation are very stable and irreversible modifications [108,109]. Since enzymes associated with peptide desulfation have not been identified in plants, it is assumed that sulfated peptides are stable signaling molecules.

Communication between cells is critical for normal plant growth and development. Receptor kinases (RKs) are the largest family of transmembrane receptors on the surface of plant cells, with 610 members in *Arabidopsis* [110]. The largest RK subfamily contains the extracellular domain of leucine rich repeats (LRR). LRR-RK mediated signaling in many cases requires somatic embryogenesis-like kinases (SERKs) as co-receptors. Five RGFR1 receptors have been identified that RGFs bind to, but with different affinities. All RGFRs 1–5 are involved in RGF signaling. However, as a result of their overlapping but unidentical expression patterns, they can play both repetitive and different roles in the regulation of plant development. This is reminiscent of CLE signaling, which has multiple peptide signals and receptors to control cell proliferation and differentiation [111].

The transcription factors PLT1 and PLT2 have been found to be proximal molecular targets through which RGF signaling occurs. The *PLTs* genes are specifically expressed in the stem cell region of the root meristem, thereby contributing to the patterning of the root stem cell niche and transient amplifying cell proliferation [112]. High levels of *PLT* support stem cells, intermediate levels promote cell division by enhancing transit, and low levels promote cell differentiation [113]. RGF signaling targets the transcription factors PLT1 and PLT2, which determine the structure of the proximal root meristem [55]. Since the expression of *PLT1* and *PLT2* and the size of the gradient are significantly reduced in the roots of *tpst-1*, but are restored after the application of the RGF peptide, it is assumed that RGF is a key factor regulating the activity of the proximal meristem through the PLT pathway [55].

Recently, two research groups have elucidated the downstream RGF1-RGFR1 signaling cascade that regulates the maintenance of the root meristem [114,115]. MITO-GEN-ACTIVATED PROTEIN KINASE 4 (MKK4) and MAP KINASE 3 (MPK3) were identified in the RGF1-RGFR1 complex [114]. Genetic and biochemical experiments have shown that MKK4 and MPK3 act as downstream signaling components of RGF1-RGFR1, modulating the expression of *PLT1* and *PLT2* [114,115] (Figure 5). It has been shown that MKK4 or MKK5 can partially repair root defects in the quintuple mutant *rgi1,2,3,4,5*. In the *mkk4.5* double mutant, the size of the root meristem was found to be significantly reduced, and *mkk4.5* plants were found to be insensitive to RGF1 treatment. Thus, it follows from these data that MKK4,5 are important participants in the RGF1-RGFR1 chain.

RGFs also play a critical role in the regulation of plant gravitropism [116,117], the formation of lateral roots and root hairs [118,119,120], and are sensitive to phosphate deprivation [82]. The results of a study using synthetic RGF peptides (DYAEPDTHPPESN, YSPAKRKPPIHN, DYKSPRHHPPRHN and, DYHSVHRHPPTHN) [121] showed that all synthetic peptides, both tyrosine sulfated and proline hydroxylated, can cause wild-type root sway in seedlings. Gravistimulation assays have shown that these peptides influence the gravitropic response of the root. Moreover, roots treated with modified RGF synthetic peptides or plants overexpressing MpRGF1 and ZmRGF1 showed a significantly increased RAM compared to the controls.

### 5.3. PLTs Regulate Stem Cell

It has recently been shown that there is a relationship between the factors PLETHORAs and WOX5, which control the content of stem cells in the *Arabidopsis* root, as well as their differentiation and proliferation [122]. The necessary longevity and continued activity of RAM can only be achieved if its pool of stem cells is continually replenished. Phytohormones and transcription factors are involved in the regulation of stem cell pool maintenance and differentiation [123]. *WOX5* is expressed mainly in QC and suppresses stem cell differentiation. The loss of *WOX5* causes the differentiation of distal stem cells, and the increased expression of *WOX5* causes their excessive proliferation. The high expression of *PLTs* in the QC region has been found to be stimulated by *WOX5*, albeit indirectly, possibly by other factors such as auxin [124]. The increased expression of *PLTs* limits WOX5 in the QC region, and the loss of PLTs is accompanied by an increase in *WOX5* and, accordingly, a large number of divisions in the QC region. Thus, there is another negative feedback loop that controls the quiescent center QC, the mutual regulation of transcription between *PLTs* and *WOX5* (Figure 6).

This regulatory loop is the missing link in the regulation of the stem cell pool by the RGF peptide, since RGF is involved in the regulation of PLT.

## 6. RALF Peptides Regulating the Root Development

### 6.1. RALF Peptides and Their Localization

Root plasticity is one of the main adaptive features that allow plants to cope with constantly changing environmental conditions. The formation and location of lateral roots along the longitudinal axis of the main root plays a vital role in the uptake of nutrients and water. Lateral roots are formed postembryonally from pericycle cells adjacent to the xylem poles [125,126,127]. Small signal peptides have been shown to be involved in the development of lateral roots [2,3,11,128]. Among the peptides involved in the regulation of lateral root development is the RALFs (Rapid ALkalinization Factor) peptide family. Some peptides, such as RALF1, RALF19, and RALF23, increase the density of emerging lateral roots, which has been demonstrated in transgenic lines [129,130] (Figure 7).

RALF was first identified in tobacco cell culture [132] as a peptide with a mol. weighing about 5 kDa. RALF is a cysteine-rich peptide containing 49 amino acid residues. The RALF peptide contains four cysteine residues, which form two disulfide bridges between the residues Cys-18 and -28 and Cys-41 and -47, which are important for the organization of the RALF structure [132]. The RALF family, which consists of more than 30 peptides, has a high homology of N-terminal amino acids in all plants [133]. The primary structure homology found in various plant species suggests that RALF plays a fundamental role in many plant families. A study of the structure and activity using the tomato RALF peptide showed that the activity requires the “YISY” motif, which is located in positions 5–8 at the N-terminus of the active peptide [134].

It was named the peptide RALF (the Rapid Alkalinization Factor) on the basis of its ability to counteract the acidification of the cell wall and the rapid increase in the level of calcium in the cytosol [132]. It was assumed that this occurs due to blockage associated with the proton pump membrane, which leads to a rapid alkalization of the medium [135]. In addition, it was shown that AtRALF1, a root-specific peptide of *Arabidopsis thaliana*, causes a temporary increase in the Ca^2+^ concentration in the cytoplasm [132].

It should be noted that most of the RALF family peptides have a negative root development. Initially, the main function of the RALF peptide was considered to be the inhibition of root elongation [128]. This conclusion was based on the results of the *AtRALF1* overexpression in *Arabidopsis*, which caused the formation of bushy, semi-dry plants with small leaves, short roots, and a decrease in the number of both lateral roots and small cells in the roots [131,136]. Conversely, the knockdown or knockout of the *AtRALF1* gene resulted in the elongation of roots and hypocotels, as well as an increase in the number of lateral roots and large root cells [131].

Based on these data, it was suggested that RALF peptides mediate a Ca^2+^–dependent signaling pathway [137]. Thus, RALF peptides play an important role in cell biology and they most likely regulate cell expansion [132,135,138,139,140].

### 6.2. RALF1 Peptide Forms Negative Feedback Loop RALF1-FER-RSL4

Most plant peptide hormones bind to leucine-rich kinase receptors (LRR-RK) located in the plasma membrane or receptor proteins (LRR-RP) [141,142]. It has been shown that RALF peptides are ligands of protein complexes, including receptor kinases Catharanthus roseus (Cr RLK1L) RLK1-LIKE [143]. It has been shown that RALF1, RALF22, and RALF23 bind to FERRONIUM Cr RLK1L (FER) to form a complex that is involved in the regulation of root growth and in response to abiotic and biotic stresses [144,145,146]. Other peptides are RALF4 and RALF19, which bind to Cr RLK1L, participating in the growth of pollen tubes and maintaining the integrity of the cell wall [147]. By binding to the kinase receptor Cr RLK1L THE-SEUS1 (THE1), the RALF34 peptide is involved in the regulation of root growth by inhibiting cellulose biosynthesis [148]. Thus, the diversity of receptors that different RALF family peptides bind to explains their functional plasticity [149].

The receptor kinase FERONIA (FER) is a universal regulator of cell growth under both normal and stress conditions [149,150]. FER binds to RALF1 and triggers a mechanism to suppress cell growth in the primary roots. In search of downstream players in RALF1-FER signaling, a receptor-like cytoplasmic kinase (RPM1-inducible protein kinase, RIPK) has been identified. RIPK interacts directly with FER and is rapidly phosphorylated upon exposure to the RALF1 peptide. RALF1 triggers FER phosphorylation and also regulates the level of phosphorylation of both FER and RIPK. Importantly, FER and RIPK phosphorylation is interdependent, indicating that not only does the upstream FER kinase activate downstream RIPK phosphorylation, but the reverse is also true. After cross-phosphorylation, FER and RIPK signal RALF1 to control cell growth in roots. FER is also involved in other processes in Arabidopsis, including the control of cell growth in leaves [151], hormonal and stress response [152,153], mechanical signaling [154], root hair development [155], and seed size control [156].

The size of root hairs is of vital physiological importance as it affects the surface area of the root and therefore the plant’s ability to absorb water and nutrients from the soil. In Arabidopsis thaliana, the extracellular RALF1 peptide and its receptor, the FERONIA receptor kinase, have been found to promote root hair (RH) growth. RALF1 promotes the phosphorylation of the EUKARYOTIC TRANSLATION INITATION FACTOR 4E1 (eIF4E1) mediated by FERONIA, a eukaryotic translation initiation factor that plays a critical role in controlling the rate of mRNA translation [150]. The phosphorylation of eIF4E1 increases affinity for FERONIA, which is accompanied by an increased affinity for mRNA and the modulation of mRNA translation, thus increased protein synthesis. mRNAs targeted by the RALF1-FERONIA-eIF4E1 module include ROP GTPase (ROP2) and ROOT HAIR DEFECTIVE 6-LIKE 4 (RSL4) [157]. The transcription factor *RSL4*, a member of a large family of *bHLH* transcription factors, regulates hair cell elongation by controlling the gene expression [158,159,160,161]. It has been noted that RSL4 regulates the *RALF1* expression in a negative feedback manner by directly binding to the *RALF1* gene promoter, thereby determining the final RH size. Thus, a negative feedback loop is formed to fine-tune the development of root hairs [150,162] (Figure 8).

It has recently been shown that the ectopic expression of *RSL4*, driven by the GLABRA2 (GL2) promoter, induces the growth of RH in *Arabidopsis* atrichoblasts, which normally do not produce RH [163]. Numerous studies have shown that RSL4 controls the expression of hundreds of genes. Together, these properties of RSL4 allow it to be identified as a major regulator of RH growth, hence the final cell size. At the same time, there are several regulatory levels, which in turn can coordinately control the transcriptional activation of *RSL4* during RH growth. Recently, more RH growth-regulating genetic components have been reported that negatively regulate the expression of *RHD6*, *RSL4*, and *RSL2* and thereby inhibit RH growth [164]. Similarly, GT-2-LIKE1 (GTL1) and its homologue DF1 bind to the *RSL4* promoter and suppress the growth of RH [165]. It is known that the phytohormone auxin is a key regulator of RH growth and induces cell growth in situ. Auxin was found to increase the *RSL4* expression by several times [162].

## 7. Molecular Mechanisms of Regulation of Root Development by Peptides

The present review considers the molecular mechanisms of regulation of the development of roots and root hairs by peptides (Table 1).

Peptides CLV3 and CLE40 bind to their receptors and suppress the *WUS* expression, which contributes to stem cell differentiation and reduction in their pool. However, the stem cell niche must be constantly replenished for normal plant development. Stem cell proliferation occurs when the activity of CLV3 and CLE40 peptides is suppressed as a result of the *WUS* activation. The formation of a negative feedback loop regulates the stem cell niche by the RGF1 peptide. However, in this variant, the main role in regulation is played not by the peptide–receptor complex, but by PLTs proteins.

The RALF1 peptide is not involved in the regulation of the stem cell niche, but it also forms a negative feedback loop to regulate root hair growth.

The review focused on the main players that form a negative feedback loop to regulate root and root hair development. However, the molecular regulatory mechanisms include a number of biomolecules that have already been identified and are yet to be identified.

## 8. Conclusions

Interest in peptide hormones has increased significantly in recent years. Peptide hormones play a variety of roles in coordinating various aspects of plant root development, including maintenance of the root meristem and the growth and formation of lateral roots and root hairs. Despite a growing body of research on the role of signal peptides in plant root development, the specific functions of many of these peptides are largely unknown. This is especially true for the mechanisms of recognition of their receptors, intracellular signaling pathways, and downstream ligand–receptor pairs. These studies are largely hampered by the fact that, having a low molecular mass, the peptides have low receptor–binding constants. The post-translational modifications of peptides increase affinity for receptors. Nevertheless, peptides are able to bind to different receptors and exhibit different regulatory functions. Therefore, the study of control over the regulation of the ligand–receptor connection seems to be the most important task in this area.

For a long time, it was believed that peptide hormones in plants only carry out intercellular signaling over short distances. However, recent studies indicate that peptides respond to environmental changes and promote plant adaptation to environmental stresses. Very important factors in the regulation of functional activity are ROS, ion fluxes and pH. Therefore, to unravel the molecular functions of peptide hormones, further comprehensive detailed studies are necessary.

## Figures and Tables

**Figure 1 plants-12-01320-f001:**
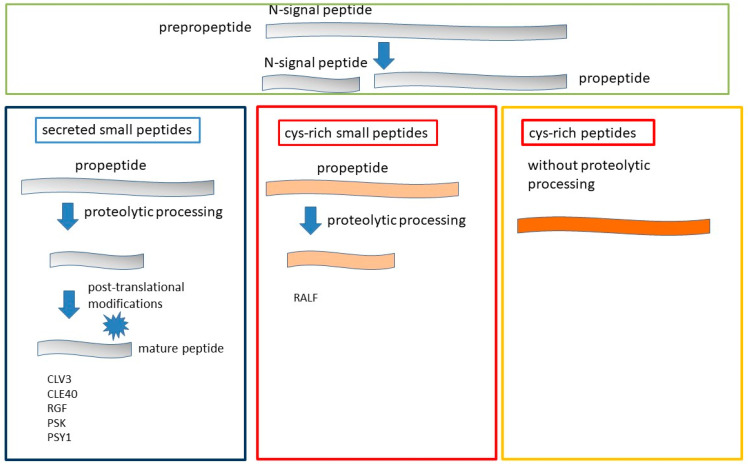
Three groups of secreted peptides classified according to their structural characteristics. The first group of peptides is small peptides with post-translational modifications. The other two groups are cis-rich peptides with and without proteolytic processing.

**Figure 2 plants-12-01320-f002:**
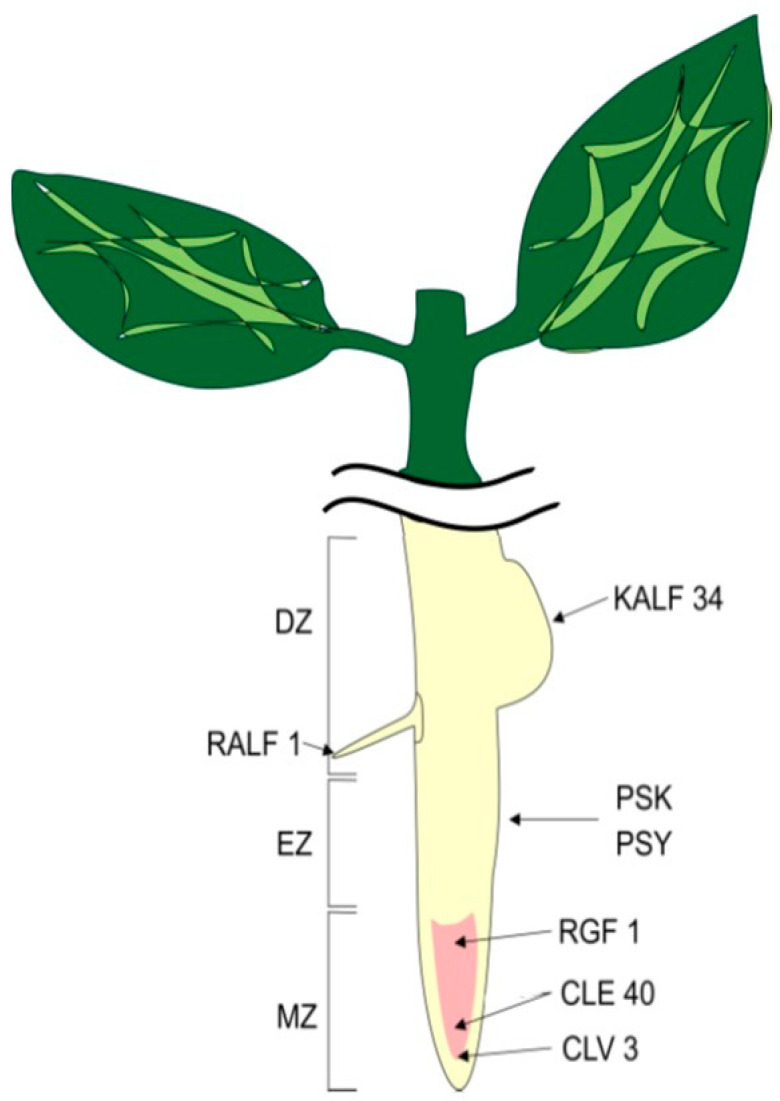
Localization of peptides in root zones. MZ—meristematic zone, EZ—elongation zone, DZ—differential zone.

**Figure 3 plants-12-01320-f003:**
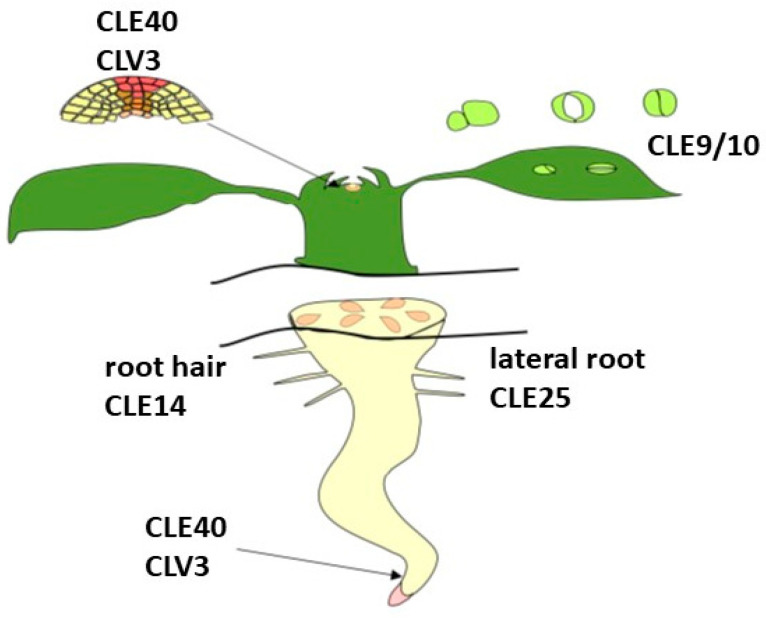
Localization CLE peptides adapted from [51].

**Figure 4 plants-12-01320-f004:**
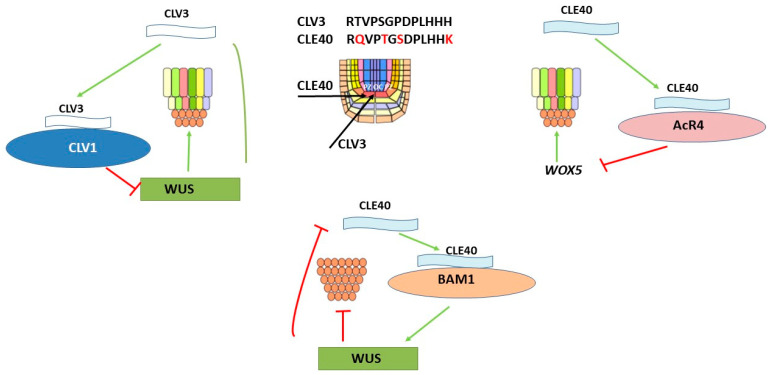
CLV3 peptide forms a negative feedback loop CLV3-CLV1-WUS in RAM; the CLE40 peptide forms a negative feedback loop CLE40-AcR4-WOX5 in RAM; the CLE40 peptide forms a negative feedback loop of CLE40-BAM1-WUS in SAM.

**Figure 5 plants-12-01320-f005:**
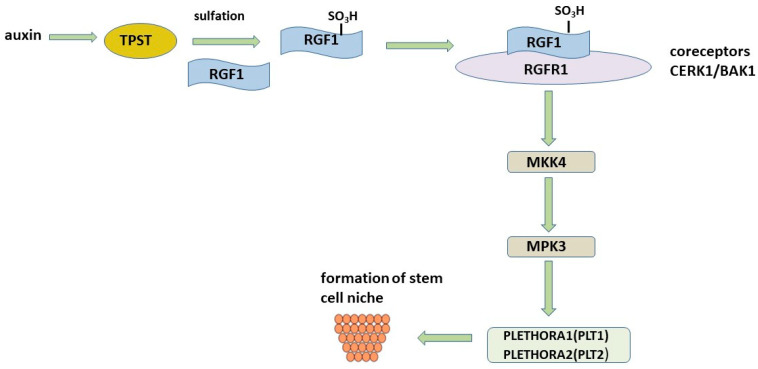
RGF stem cell niche formation pathway.

**Figure 6 plants-12-01320-f006:**
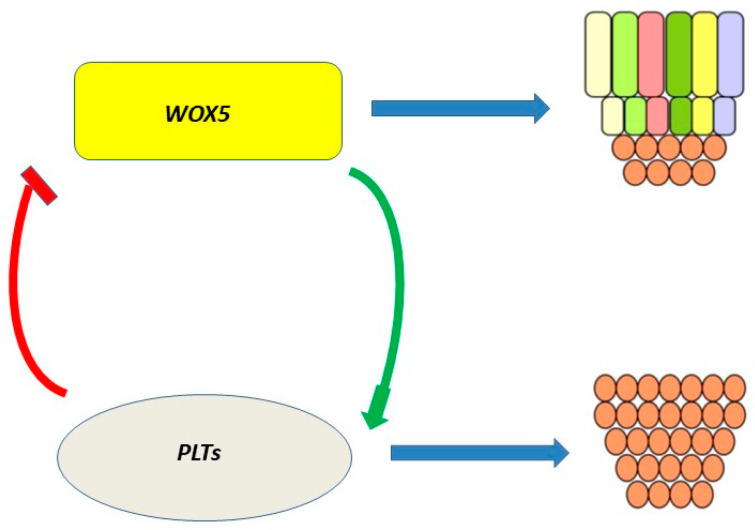
Relationship between *PLTs* and *WOX5*. High *WOX5* expression causes stem cell proliferation, and its loss causes their differentiation.

**Figure 7 plants-12-01320-f007:**
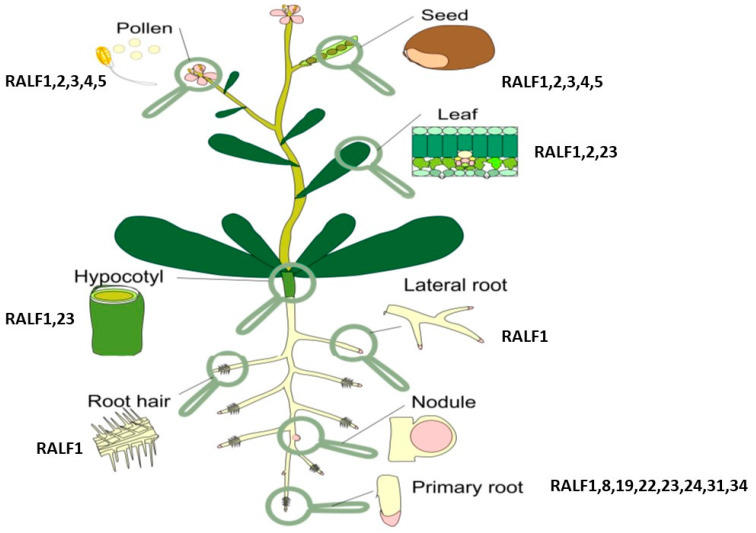
Localization of RALF peptides adapted from [131].

**Figure 8 plants-12-01320-f008:**
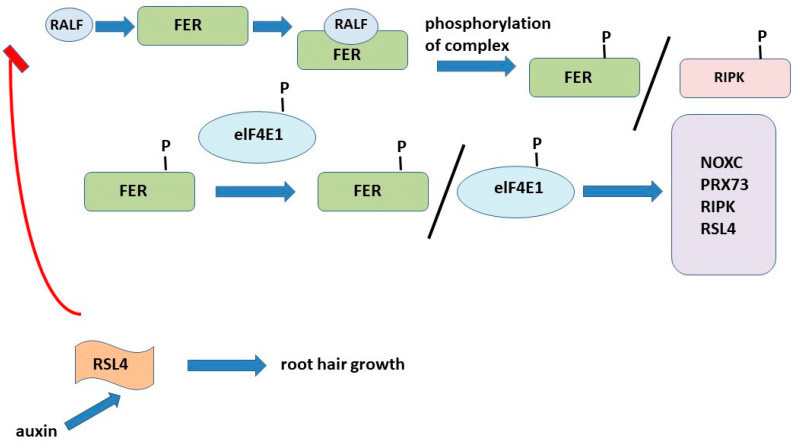
Negative feedback loop RALF1-FER-RSL4.

**Table 1 plants-12-01320-t001:** Molecular mechanisms of regulation of root development by peptides.

Peptide Receptor Negative Feedback Loop Regulatory Function
CLV3 [49] CLV1 [25] CLV3-CLV1-WUS in RAM [48,61] stem cells niche
CLV3 [49] CLV1 [25] CLV3-CLV1-WUS in SAM [75] stem cells niche
CLE40 [24] AcR4 [82] CLE40-AcR4-WOX5 in RAM [83] stem cells niche
CLE40 [24] BAM1 [85] CLE40-BAM1-WUS in SAM [89] stem cells niche
RGF1 [55] RGFR1 [104] RGF1-RGFR1-PLTs-WOX5 [122] stem cells niche
RALF1 [137] FERONIA [144] RALF1-FER-RLS4 [161] root hair growth

## Data Availability

The data presented in this study are available on request from the corresponding author.

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
