# Peer review of "Molecular Mechanisms of Regulation of Root Development by Plant Peptides"

_plants, 2023, doi:10.3390/plants12061320_

Round 1
Reviewer 1 Report
The ms sounds very interesting but some corrections should be done before being being accepted to be publised.
1. I suggest using the same terms to indicate "molecular weigh" along the main text.
2. Although similar, I find most frequently the term "proteases" than proteinases.
3. Please, consider adding an abbreviations list, otherwise introduce each abbreviature when mentioned first time.
4. The final paragraph in the introduction should include some references, as I deduce they are not sentences based on your own work.
5. Lines 98-99 are confused.
6. Line 168. Refer you are talking about SAM
7. Line 187. Refer when saying ..."QC or resting center" you are talking about RAM
8. Line 191. Replace by CLV3-CLV1-2 at the begining of legend.
9. Line 205. Reference 68 is numbering wrong. Please, revise. It is expected 66.
10. Lines 243 and 246. Do not use QC but OC as it is referring to SAM.
11. Lines 311-312 and 320-321 repeat the same thought. Also it hapens with the lines 424-425 and 429-430.
12. Line 486. Replace by "in nutrient and water uptake"
13. Line 489. ...among them, the RALFs...
14. Line 573. Phosphorilated...polarity...is confused. Please, explain.
15. Please, rewording of some paragraphs is needed:
Pps: 82-86; 140-149; 155-157; 263-264; 308-311; 339-355; 362-365; 386-387; 436-443; 609-623; 631-632.
16. There are several mistakes with the list of references. Please, pay attention to underlines and numbering.
Author Response
Thank you very much for your review and valuable comments. All comments are taken into account and added to the text.

Reviewer 2 Report
the manuscript is a review of the peptides in plants and their role in root development. The introduction is well-written and reports an overview of the plant peptides. The main references have been reported. Appropriate old and recent references have been considered. The classification and regulatory function of the peptides are well organised. The draws reported are very nice and useful. I would suggest revising the text considering the following suggestions:
-please write in italics the gene names.
-row 159, write the acronym CLE the first time that you cite it
-row 171, please explain in which way CLEs respond to the water and nitrogen, they are just mentioned.
-legends of figures must be self-explanatory, please improve them.
Author Response

(The authors gave the same response as above.)

Reviewer 3 Report
This Review contains actual, important and detailed information about peptides in root development. I consider the paper with good potential for publication in Plants.
Some suggestions of improvements:
Abstract: Should be rewritten with more information about the main findings of the review
Key-words: Should be complete reviewed
Introduction: see commentaries in the PDF version (attached)
Whole text: There are a lot of information of several aspects of plant development in response to different type of peptides. Although looks positive, the large quantity of text and the too much specific information contained resulted in a cansative, confuse and few readable manuscript.
To improve this part, I suggest authors to increase subtitlesand better organize the text, according some aspect (peptide, effect in the part of roots, etc). Also, due the large quantity of text, I recommend authors to construct at least one table containing detailed information about each study, as a way to organize the information about each specific study realized with peptides and root development). These two suggestions could help readers to understood the state of art and to find the main studies with specific applications. Also, will allow authors to reduce the whole text to the main finds resulted from the review, turning the text more readable. In general, the main findings obtained for each authors could be resumed in a table, and authors could write more about the main findings in this area.

Author Response

(The authors gave the same response as above.)

Reviewer 4 Report
This review is written around plant peptides, it would give the readers a holistic picture about types of peptides if the authors could add one or two sentences in the first paragraph regarding different sources of peptides with citations.
Minor revisions are suggested as below:
Line Number 3 Since this review focuses on the effect of peptide hormones on the root development, the “peptides” in title may be changed to “peptide hormones”
9 more specifically, “class” of what?
20 need to insert an abbreviation list here
32 “proteins” be changed to “protein”
37-39 need to rephrase this sentence
46 “AMPs are encoded by genes” ==> AMP genes
50 “another”==> the other
68 “were” ==> are
153 first abbr. of CLE need explanation
154 first abbr. of CLV need explanation
200 Why QC is used for “calm center”
394-396 need to rephrase this sentence
399 “for sulfation”==>of sulfation
403 “”==>increase
406 second “that”==> the
410 “members of”==>members in
Author Response

(The authors gave the same response as above.)

Round 2
Reviewer 3 Report
I continue to understand the large importance of the paper and their suitability for this journal. In response to review realized in first version, authors justified the most of recommendations realized in their paper and not corrected sufficiently their paper, mainly the structure that continues difficult for readers. In addition, some paragraphs are repeated more than one time in the text, showing that this large quantity of information confuse author and readers (as example, see the last paragraph added to conclusion - this paragraph was repeated two times in the same conclusion section -659-660 and 668-670). Also there are minor errors along the text that is not completely corrected by author and should be revised.
The paper are strong, with high quality information and good figures for publication, but I continue with my suggestion that should be better organized as I asked in the first review.
Author Response
Thank you for your review and for your valuable advice. According to your suggestions, the sections were divided into subsections, the main theses of the review were presented in a table in a separate section. Your comments on the text have been corrected. Sorry if not all of your comments were taken into account.

Round 3
Reviewer 3 Report
Dear Authors, thanks for your reviwed manuscript. The text now are improved and already for publication. This division into subtopics easy for the readers that will find for specific characteristics from the effects of peptides on plant development. Congratulations for your rich review about peptides in plants.
Author Response
Thank you very much for the positive review. Indeed, after the revision of the material, the manuscript is much more attractive.